# Loose lips sink ships: Mitigating Length Bias in Reinforcement Learning from Human Feedback

**Wei Shen**[1*], **Rui Zheng**[1*], **Wenyu Zhan**[1], **Jun Zhao**[1], **Shihan Dou**[1],
**Tao Gui**[3†], **Qi Zhang**[1], **Xuanjing Huang**[1,2†]

[1] School of Computer Science, Fudan University
[2] International Human Phenome Institutes (Shanghai), Shanghai, China
[3] Institute of Modern Languages and Linguistics, Fudan University
wshen21@m.fudan.edu.cn,
{ruizheng20,tgui,xjhuang}@fudan.edu.cn

## Abstract

Reinforcement learning from human feedback serves as a crucial bridge, aligning large language models with human and societal values. This alignment requires a vast corpus of human feedback to learn a reward model, which is subsequently used to finetune language models. However, we have identified that the reward model often finds shortcuts to bypass its intended objectives, misleadingly assuming that humans prefer longer responses. The emergence of length bias often induces the model to favor longer outputs, yet it doesn't equate to an increase in helpful information within these outputs. In this paper, we propose an innovative solution, applying the Product-of-Experts (PoE) technique to separate reward modeling from the influence of sequence length. In our framework, the main expert concentrates on understanding human intents, while the biased expert targets the identification and capture of length bias. To further enhance the learning of bias, we introduce perturbations into the bias-focused expert, disrupting the flow of semantic information. Experimental results validate the effectiveness of our approach, indicating that language model performance is improved, irrespective of sequence length.

## 1 Introduction

In recent years, the field of natural language processing has witnessed remarkable advancements with the emergence of powerful models like InstructGPT (Ouyang et al., 2022), GPT-4 (OpenAI, 2023), Claude (Bai et al., 2022b), and others. These models have displayed remarkable proficiency in understanding human queries and providing helpful responses. Their success can be attributed to a two-step learning process that involves Supervised Fine-Tuning (SFT) followed by the utilization of Reinforcement Learning from

---

*Equal contribution.
†Corresponding author.

Figure 1: Comparison of model outputs using SFT, Vanilla RM with PPO, and Our Method with PPO. Our study demonstrates that the current reward model tends to incentivize the model to generate longer responses, disregarding the true human intent. This phenomenon is highlighted in the example shown, where we observe a decline in model performance.

Human Feedback (RLHF) techniques (Ouyang et al., 2022; Bai et al., 2022a). This combination enables these models to not only learn how to follow human instructions but also better understand human intent and align with human and societal values. Undoubtedly, RLHF plays a pivotal role in the success of these models.

One of the key components of RLHF is reward modeling (RM), which involves learning a reward function from human preferences or demonstrations. This allows an RL agent to optimize its behavior based on the feedback received from reward model (Ziegler et al., 2019b). However, the process is not without challenges. Human preference data can be noisy and subject to inconsistencies among different annotators, leading to suboptimal results (Bai et al., 2022a).

For example, reward gaming, a well-known and pervasive issue, refers to the phenomenon where trained models exhibit undesirable patterns and generate low-quality outputs while still receiving high rewards (Skalse et al., 2022; Pan et al., 2022). These complexities emphasize the need for careful consideration and robust methods to ensure reliable and meaningful reward functions in RLHF.

As shown in Figure 1, a similar reward gaming issue arises in NLP reward modeling. We have observed that the reward model tends to reply on simple patterns, such as sentence length, to differentiate between good and bad responses. Typically, the reward model assumes that longer responses are better, which hinders its ability to learn the true human intent and preference. Addressing this problem is crucial for improving the effectiveness of NLP reward modeling and capturing the true nuances of human language.

In this paper, we propose a Product-of-Experts (PoE)-based method (Hinton, 2002) that consists of two expert models to decouple human intent and response length during the reward modeling phase. The first expert operates similarly to a standard reward model and focuses on learning the true human intent behind responses. The second expert, referred to as the bias-only expert, is designed to learn simple patterns, specifically the length of responses. It employs a smaller model capacity and a larger learning rate to capture coarse-grained information of inputs. Additionally, stochastic perturbations are introduced into the inputs of the bias-only expert, intentionally disrupting the semantic information present in the input. To summarize, the main contributions of our work are followings:

- We identify that reward modeling in NLP tends to rely on length bias, hindering the models from accurately learning true human intent and even leading to model degradation.

- We propose a simple and efficient solution leveraging PoE technique. Our method effectively decouples length bias from human intent, enabling models to better capture and understand human preferences.

- We validate the effectiveness of our proposed method. The results show that our approach enhances the learning of human intent by avoiding the generation of meaningless and overly verbose outputs.

## 2 Related Work

**Reinforcement Learning from Human Feedback.** Using human preference feedback is a popular way for realizing AI alignment (Leike et al., 2018). Preferences are often provide numerical value or demonstrations (WirthChristian et al., 2017) without requiring expert proficiency or fine-grained feedback . Alignment bring a potent capability to state-of–the-art generative foundation models, like InstructGPT (Ouyang et al., 2022), Sparrow (Glaese et al., 2022), Claude (Bai et al., 2022b), which means this method is of great success in the paradigm of learning with human feedback. Some prior work have explored using human feedback to improve various tasks, such as summarization (Stiennon et al., 2022; Ziegler et al., 2019b), dialogue (Bai et al., 2022a), translation (Bahdanau et al., 2016), event generation (Zhou and Xu, 2020), semantic parsing (Lawrence and Riezler, 2019) and instruction following (Ouyang et al., 2022; Bai et al., 2022a). These work can be categoried as supervised fine-tuing or reward modeling on well-constructed human annotations information, the latter is also known as a vital phase in RL from human feedback (Christiano et al., 2023; MacGlashan et al., 2017). Our work falls within the realm of RLHF and aims to awaken LM with both harmless and helpful abilities.

**Reward Hacking.** Goodhart's Law[1] (Strathern, 1997) can be formulated a tough challenge in numerous fields. A few approaches have been proposed for reducing overoptimization in general reinforcement learning (Everitt et al., 2017), as well as in reward models (Gleave and Irving, 2022). The overoptimization problem of reward model can generally regard as a special case of reward gaming, also known as reward hacking (Skalse et al., 2022). In addition, Pan et al. (2022) proposed to systematically analyze reward misspecification in RL by creating a set of domains where the agent optimizes a hand-engineered proxy reward function. In this study, we consider the length of human preferences as a confounding factor that hinders the reward model from accurately assessing the quality of model responses based on true human intent.

**Products-of-Experts.** Products-of-Experts (PoE) (Hinton, 2002) has been proposed as an alternative to mixture model to compensate for their poor

---

[1]Goodhart's law is an adage often stated as, *When a measure becomes a target, it ceases to be a good measure*

efficiency in high dimensional space. This technique is often based on the principle of *wisdom of crowds*, which suggests that aggregating multiple models can lead to better performance than relying on a single model. Clark et al. (2019) firstly use PoE to build a paradigm that train a debiased model ensemble with a bias-only model. The goal is to encourage the debiased model to utilize orthogonal information with information from the bias-only model. Typically, this kind of method (Clark et al., 2019; He et al., 2019) usually contains two stages. In this work, we adopt the end-to-end manner like (Karimi Mahabadi et al., 2020) which jointly learn the bias-only model and the debiased main model simultaneously, therefore, our reward model can take advantage of PoE to use a weak learner to capture the length shortcuts without any prior information about the length of sentences, and the main model can purely attain the correct knowledge that is suitable for human preference.

## 3 Preliminary

For the purpose of implement RLHF, we follow the pipeline in Ziegler et al. (2019a). It is usually made up of three phrases: 1) supervised fine-tuning, 2) reward modeling, 3) reinforcement-learning optimization, our attention is directed towards the last two phases.

**SFT:** It begins with a generic pre-trained LM, which is fine-tuned using supervised learning on a high-quality instruction dataset. This allows the model to follow various instructions, perform dialogue and dialogue. As a result, we obtain an LM $\pi^{\text{SFT}}$ during this phase.

**Reward Modeling:** According to the Bradley-Terry model (Bradley and Terry, 1952), a reward function is hard to describe and needs to be learned from preferences among trajectories. The reward model $r_\theta$ is trained on human preference dataset to predict which response $y \in \{0, 1\}$ is better as judged by human, given a query $x$. If the response preferred by human is $y_i$, the RM loss can be expressed as:

$$-\mathbb{E}_{(x,y)\sim\mathcal{D}}[\log(\sigma(r_\theta(x, y_i) - r_\theta(x, y_{1-i})))], \quad (1)$$

where $r_\theta(x, y)$ is the scalar output of the reward model for query $x$ and response $y$ with parameters $\theta$, and $\mathcal{D}$ is the human preference dataset.

**RL Optimization:** Then we fine-tune the SFT model on our environment using PPO (Schulman et al., 2017). The language model is provided with

feedback through the launch of a learned reward function. To this end, we maximize the following objective function in RL training:

$$\mathbb{E}_{(x,y)\sim\mathcal{D}}\left[r_\theta(x, y) - \beta \log\left(\pi_\phi^{\text{RL}}(y \mid x)/\pi^{\text{SFT}}(y \mid x)\right)\right],$$

$\beta$ is a parameter to control the deviation from the SFT model $\pi^{\text{SFT}}$. The language model policy $\pi_\phi$ is initialized to $\pi^{\text{SFT}}$. Importantly, the last per-token KL-penalty term is used to prevent the model from deviating too far from the deviating exceeding the appropriate scope from the distribution on which the reward model is accurate, as well as maintaining the generation diversity and preventing mode-collapse to single high-reward answers (Ziegler et al., 2019b).

## 4 Length Bias in Reward Model

In this section, we present the phenomenon of length bias in reward models and utilize causal analysis to examine the underlying reasons for this occurrence. We shed light on why the reward model exhibits a preference for longer sentences and delve into the causal factors contributing to this bias.

### 4.1 Length Bias Phenomenon

We present a Figure 2 depicting the scores and lengths of 4000 SFT model output results, which were evaluated using the vanilla reward model trained on the helpful and harmless (HH) dataset (Bai et al., 2022b). It is evident that there is a strong correlation between the reward scores and lengths. When the model generates longer sequences, the reward model tends to assign higher scores. This correlation contradicts human intent since the helpfulness and harmlessness of the output should not be solely determined by its length. More figures can be seen in the Appendix 8.

In addition, the length bias locates in PPO as well, we additionally investigate it on TL;DR (Stiennon et al., 2022) in the Appendix A.4

### 4.2 Confounding Factor

As Figure 3 depicted, we formulate the problem as a causal structure (Zeng et al., 2023; Tien et al., 2023) of preference-based reward modeling. Pairwise preference information are conveyed based on an observed reward function $r$. Features $(x, y)$ are causal factors can affect $r$ and another nuisance features $z$ regarded as a potential confounder factor that have impact on $r$. Note that

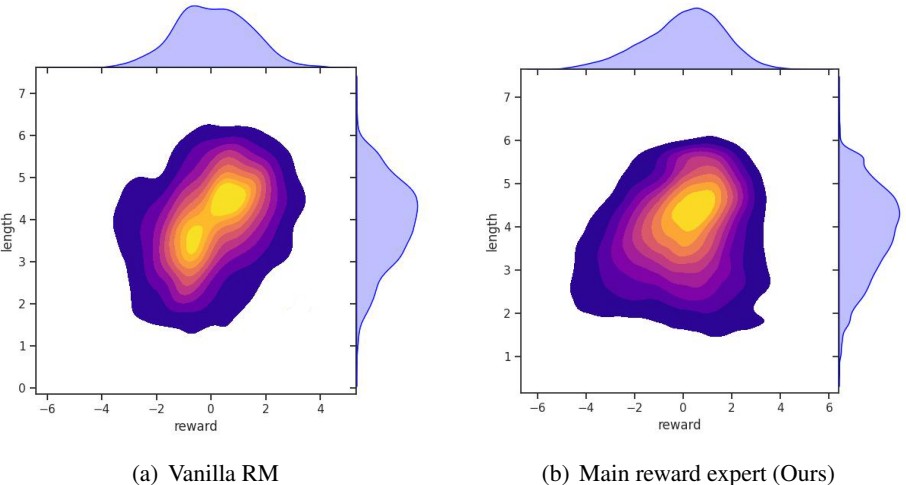

(a) Vanilla RM  (b) Main reward expert (Ours)

Figure 2: Distributions of Reward vs. log-scaled Length in vanilla RM and our proposed method. The results illustrate that Vanilla RM tends to assign higher rewards to longer responses, whereas our proposed method achieves a more uniform distribution of rewards across different response lengths.

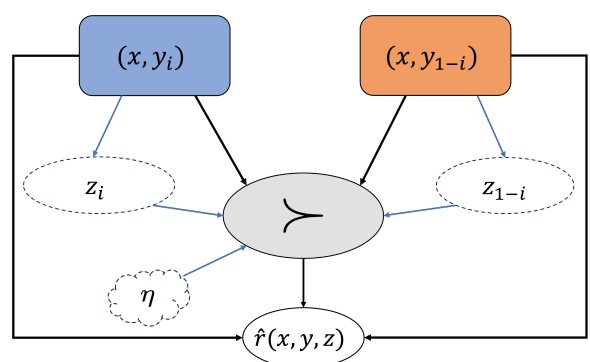

Figure 3: Causal structure of preference-based reward modeling. Length bias, represented as a confounding factor $z$, influences the model's learning of true human preferences $(x, y)$.

$z$ might be correlated with $(x, y)$, potentially due to their sharing of the same causal parent or there being biases during data collection, like annotators are inclined to favor longer sentences, which is expected.

In the presence of confounding effects, the goal of preference-based reward modeling is to learn a biased reward function $\hat{r}(x, y, z)$ that best matches the stated preferences. There are state features $(x, y)$ that are causal with respect to the true human intent but are unobserved by the learning agent and considered non-robust features effortlessly learned. In our case, we suppose that length bias is one vital element strongly related to $(x, y)$. Actually, this situation can hardly be completely avoided, and our motivation is to reduce the probability of the path directed by the blue arrow. Furthermore, unobserved human bias and noise induced by annotators, such as inappropriate judgments on certain individuals, groups, races, sexes, etc., denoted by $\eta$, also affect the preference labels.

As a consequence, the learned reward $\hat{r}$ has risk to achieve actually low yet even near-perfect performance on a held-out test dataset with following two factors: 1) causal factor and nuisance factor can be correlated by the con-founder factor $(x, y)$, 2) RM easily find a shortcut exists between causal factor and reward function in the distribution of test set and can not be extrapolated to out-of-domain dataset. This might be reflected when this misspecificated reward model $\hat{r}(x, y, z)$ guide the LM agent to align with human intention using PPO, as a result of distribution drift.

### 4.3 Challenges

The spurious correlation between reward scalars and response lengths can be problematic, as it may result in biased or sub-optimal model behavior. However, due to the unobserved nature of this spurious attribute when performing a traditional ERM-based training paradigm, it can be challenging to investigate and address during the reward modeling stage. Inspired by the debiasing framework of ensemble model methods, we posit that length bias can be alleviated through a robust learning approach, which involves disentangling the underlying features and feeding them into distinct experts. This assumption is grounded

in the notion that the bias in the length of data can be attributed to the confounding effect of certain features, which can be disentangled through representation learning. Based on the aforementioned observations and insights, in the next section, we will propose a simple and effective method to mitigate the length bias during the reward modeling stage.

# 5 Proposed Method

The section describes the framework and algorithm of our method. We introduce an approach to establish a debias framework that can significantly mitigate the length bias at the RLHF stage.

## 5.1 PoE Framework

In this study, our framework is mainly built based on the procedure of the reward modeling phase, as illustrated in Figure 4. To learn a reward model from preferences, we assume access to a set of pairwise preference pairs. Specifically, after a batch of $N$ data consisting of an equal quantity of positive and negative sentences passes through our framework, the learning process begins.

**Products-of-Experts.** In our study, we explore a reward modeling method that utilizes different experts to separately learn the true human intent and the length bias. We employ the Products-of-Experts (Hinton, 2002) technique to train a robust reward model. Specifically, our ensemble method can be formulated as:

$$\hat{r}(x,y) = \text{Softmax}(\log(r_\phi(x,y)) + \log(r_\psi(x,y))).$$

Equivalently, $\hat{r}(x,y) \propto r_\phi(x,y) \circ r_\psi(x,y)$, where $r_\psi(x,y)$ is the output of the bias-only model and $r_\phi(x,y)$ is that of the main reward model.

To ensure the main reward expert and bias-only reward expert learn different content, we applied constraints based on empirical observations. The main reward model utilized a larger language expert (e.g., 7B LLAMA (Touvron et al., 2023)) and a normal learning rate to capture human intent. In contrast, the bias-focused expert employed a smaller model (e.g., 560M BLOOMZ (Muennighoff et al., 2022)) and a higher learning rate, typically three times that of the main expert. Previous studies (Mhaskar et al., 2016; Wilson et al., 2018) showed that smaller models with larger learning rates tend to learn simpler and coarser information. This differentiation in model size and learning rate aims to balance comprehensive understanding with the identification and mitigation of potential biases (Geirhos et al., 2020).

## 5.2 Injecting Noise into Bias-only Expert

In order to ensure that the bias-only model captures the length bias present in the input, we employ a technique of injecting random noise into the input. This intentional introduction of noise serves the purpose of disrupting the semantic information within the input, thereby aiding the model in effectively learning the length bias. Thus, the perturbed inputs can be expressed as $X' = X + N$, where $X'$ represents the new, noisy input, and $N$ denotes the Gaussian noise added to the token embeddings. By facilitating the collaboration between bias-only experts and main experts in modeling human preference, we enable the bias-only experts to effectively capture length bias while preventing the main model from learning length bias.

## 5.3 Training & Inference

During the training phase, the main expert and the bias-only expert are jointly optimized by maximizing the following likelihood function to optimize the reward function:

$$-\mathbb{E}_{(x,y)\sim\mathcal{D}}[\log(\sigma(\hat{r}(x,y_i) - \hat{r}(x,y_{1-i})))]. \quad (2)$$

The main expert is initialized based on an SFT model, while the bias-only expert is initialized using a pretrained model. Both reward models add a linear layer on top of the final Transformer layer, which generates the final scalar prediction for the reward signal.

During the PPO stage, we exclusively rely on the main expert to provide rewards, while discarding the bias-only expert. Since the bias-only expert is typically smaller in size, our approach does not significantly increase computational overhead.

# 6 Experiments

## 6.1 Settings

**Datasets** We utilize the Helpful and Harmless (HH) dataset (Bai et al., 2022b) from Anthropic as our experimental dataset and rm-static[2] for training our reward model and for participation in PPO. The HH dataset provides a response

---

[2]https://huggingface.co/datasets/Dahoas/rm-static

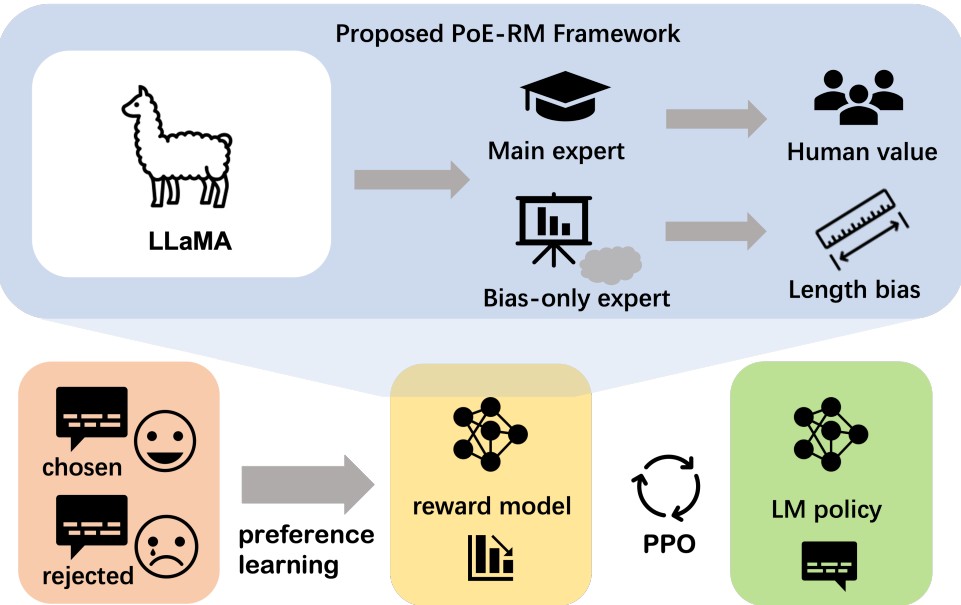

Figure 4: Pipeline of the proposed method. The proposed PoE-based reward modeling approach consists of two experts: the main expert, which learns the true human value, and the bias-only expert, which focuses on capturing the length bias.

and a rejected response for each query based on human preferences, specifically focusing on responses that are helpful and harmless. In addition to that, our SFT data incorporated the 52k instruction dataset constructed by Alpaca[3] and the ChatAlpaca[4] dataset containing multi-turn dialogues.

**Models** In our experimental setup, we primarily build upon LLaMA and BLOOMZ models, utilizing models with a parameter size of 7B. Inspired by the work of (Ouyang et al., 2022), who employ SFT models as initial models for PPO, we perform SFT on the Alpaca and ChatAlpaca datasets.

**SFT Hyper-parameters** During the SFT phase, we utilize a learning rate of $3e^{-5}$ and train for three epochs without early stopping. We employ a warmup period of $0.3$ epochs, followed by a linear decay to $0$. The fine-tuning process was conducted on a device with eight Nvidia A100 GPUs. Each GPU handled four queries, resulting in a batch size of $32$. Responses are truncated to $512$ tokens, while the total length of both queries and responses was truncated to $2048$ tokens. We incorporate specific prompts, such as "Human:" or "Assistant:", during the concatenation of input queries and output responses. These prompts are added to provide

context and distinguish between human-generated and assistant-generated responses.

**RLHF Hyper-parameters** During the reward modeling training phase, our main expert and policy model remained consistent. The learning rate for both is set to $5e^{-6}$. As for the bias-only expert, we utilize a smaller model, the 560m Bloomz, with a fixed learning rate of $8e^{-6}$.

In the PPO framework, we perform reward score normalization and clipping, with a clip value of $0.8$. We employ the clipped surrogate objective of PPO for optimization. The token-level KL penalty coefficient $\beta$ is set to $0.05$. For each query, we collect $4$ roll-out samples using nucleus sampling. The sampling temperature is set to $0.8$, top-p is set to $0.9$, repetition penalty is set to $1.1$, and the maximum output token length is set to $512$. The policy model has a learning rate of $9e^{-7}$, while the value model utilize a learning rate of $2e^{-6}$. These specific training details are implemented to optimize the performance and convergence of the models during the training process.

**Baselines** In this study, we propose a method primarily aimed at mitigating length bias in the reward model. Therefore, our baselines include the SFT model, the PPO model trained with the vanilla reward model, and the PPO model trained exclusively with the bias-only reward expert.

---

[3]https://github.com/tatsu-lab/stanford_alpaca
[4]https://github.com/cascip/ChatAlpaca

| Dataset | Models | RM Type | | | | | Length | PPL |
|---------|--------|---------|--------|-----------|--------------|-------|--------|-----|
| | | LLaMA | BLOOMZ | Ours-LLaMA | Ours-BLOOMZ | GPT-J | | |
| **HH-RLHF** | BLOOMZ-SFT | 0.986 | −1.008 | 1.098 | 0.285 | 0.162 | 502 | 10.09 |
| | Vanilla-PPO-BLOOMZ | 1,082 | −1.161 | 1.183 | −0.422 | 0.174 | 513 | 10.06 |
| | Ours-PPO-BLOOMZ | 1.867 | −1.162 | 1.313 | −0.327 | 0.167 | **507** | **9.13** |
| | LLaMA-SFT | 1.386 | −1.179 | 1.854 | 0.354 | 0.174 | 468 | 10.16 |
| | Vanilla-PPO-LLaMA | 1.580 | −0.997 | 2.106 | −0.200 | 0.196 | 503 | 9.81 |
| | Ours-PPO-LLaMA | 1.474 | −1.062 | 1.975 | −0.256 | 0.172 | **485** | **8.92** |
| | Alpaca-SFT | 1.557 | −1.008 | 2.152 | 0.413 | 0.191 | 513 | 10.11 |
| | Vanilla-PPO-Alpaca | 2.052 | −0.623 | 2.882 | 0.152 | 0.241 | 689 | 9.67 |
| | Ours-PPO-Alpaca | 2.041 | −0.587 | 2.850 | 0.150 | 0.221 | **586** | **8.82** |
| | Biasd-PPO-Alpaca | 1.948 | −0.604 | 2.930 | 0.025 | 0.148 | 684 | 9.91 |
| **rm-static** | BLOOMZ-SFT | 1.463 | 0.101 | 1.315 | 0.293 | 0.164 | 494 | 10.03 |
| | Vanilla-PPO-BLOOMZ | 1.599 | 0.242 | 1.480 | 0.421 | 0.241 | 517 | 9.98 |
| | Ours-PPO-BLOOMZ | 1.688 | 0.324 | 1.603 | 0.487 | 0.239 | **501** | **8.84** |
| | LLaMA-SFT | 2.026 | 0.365 | 2.367 | 0.530 | 0.182 | 473 | 9.53 |
| | Vanilla-PPO-LLaMA | 2.101 | 0.463 | 2.480 | 0.599 | 0.262 | 513 | 9.30 |
| | Ours-PPO-LLaMA | 2.045 | 0.430 | 2.464 | 0.585 | 0.249 | **489** | **8.89** |
| | Alpaca-SFT | 2.224 | 0.597 | 2.807 | 0.736 | 0.193 | 532 | 9.53 |
| | Vanilla-PPO-Alpaca | 2.563 | 0.931 | 3.414 | 0.931 | 0.291 | 721 | 9.03 |
| | Ours-PPO-Alpaca | 2.587 | 0.919 | 3.428 | 0.968 | 0.289 | **617** | **8.80** |
| | Biasd-PPO-Alpaca | 2.848 | 1.260 | 3.845 | 1.232 | 0.152 | 713 | 9.57 |

Table 1: Main result for our proposed framework, we trained multiple models following RLHF pipeline. And we use five different Reward Model to comprehensively evaluate the sentences against different settings, which are generated from model-generated 4608 and 2304 prompts extracted from hh-rlhf and rm-static respectively.

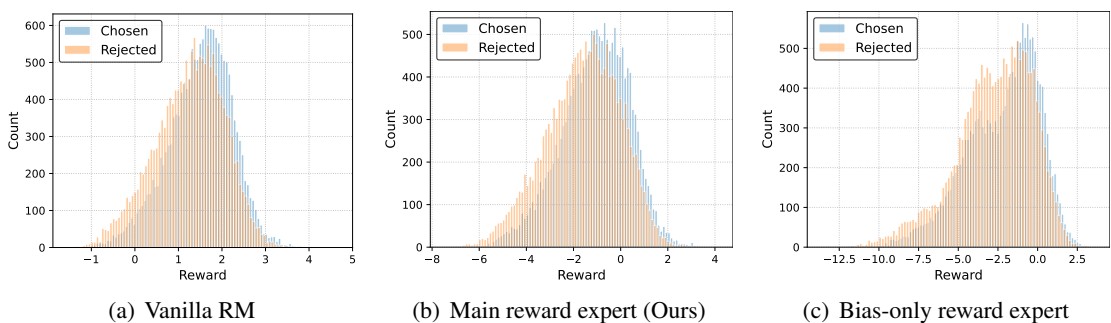

(a) Vanilla RM      (b) Main reward expert (Ours)      (c) Bias-only reward expert

Figure 5: Distribution of reward scores for chosen and rejected responses. The results demonstrate that our proposed method increases the differentiation between chosen and rejected responses. In contrast, the bias-only expert exhibits limited generalization ability, primarily relying on length bias.

| Models | RM % | Bias-only % |
|--------|------|-------------|
| Vanilla-BLOOMZ | 66.94 | − |
| **w/o** PoE | 67.11 | 66.73 |
| **w/o** Input Noise | **67.53** | 66.58 |

Table 2: Ablation study on HH dataset, RM Accuracy and bias-only RM for BLOOMZ models. By employing a combination of PoE and input noise perturbation on enhancing the generalization capability of the reward model. The performance on the test set significantly improves as a result of these techniques

**Metrics** We evaluate the effectiveness of different methods in our experiments using perplexity (gpt2-medium), average reward score , and human

evaluators. For human evaluation, annotators compared two randomly selected responses and provided comparison results (win/lose/tie), allowing us to gain insights into subjective judgments of response quality.

## 6.2 True Reward Improvement

As the previous analysis, addressing the risk of reward hacking resulting from reward model overoptimization (Gao et al., 2022) is crucial. To ensure a more robust evaluation of our method, we adopted a comprehensive approach that combines both automatic and human evaluation. Table 1 illustrates the average generated sequence length and reward scores of our method in comparison to

the baselines on the test set. The results clearly demonstrate that our approach achieves higher reward scores while simultaneously reducing the average output length compared to the RL model utilizing the vanilla reward model. This finding not only confirms the effectiveness of our proposed method but also establishes a foundation for generating outputs that better align with human preferences.

In addition, we conducted a Spearman/Pearson analysis to further validate the effectiveness of our method. For more detailed information, please refer to Appendix 4.

| | Opponent | Human | AlpacaFarm | GPT-4 |
|---|---|---|---|---|
| Ours | Answer$_{\text{Chosen}}$ | 67.69 | 74.42 | 75.32 |
| Ours | SFT | 54.23 | 59.42 | 56.42 |
| Ours | PPO | 57.47 | 61.43 | 59.56 |

Table 3: Win rates of our proposed method against various baseline approaches in the HH-RLHF dataset, utilizing the Alpaca. Evaluations include human annotators, AlpcaFarm and GPT-4. Answer$_{\text{Chosen}}$ is the human preferred response in HH-RLHF.

### 6.3 Wining Rate

In this section of the experimental analysis, we present the win rate of our method compared to other approaches. We provide results from both manual evaluations, GPT4 and the automated evaluation platform, and AlpacaFarm. During the pairwise comparisons of the model results, a win is assigned $1$ point, a tie is assigned $0.5$ points, and a loss is assigned $0$ points. It is evident that our method achieves an average win rate of more that $50\%$ compared to traditional RLHF methods. This validates the effectiveness of our proposed method in generating outputs that are more informative and concise.

### 6.4 Ablation Study

Table 2 presents the results of ablation analysis on various components of our method. By utilizing the PoE technique and input perturbation, we observe an improvement in the accuracy of the RM. This finding demonstrates that our method can better identify human intent, leading to enhanced performance.

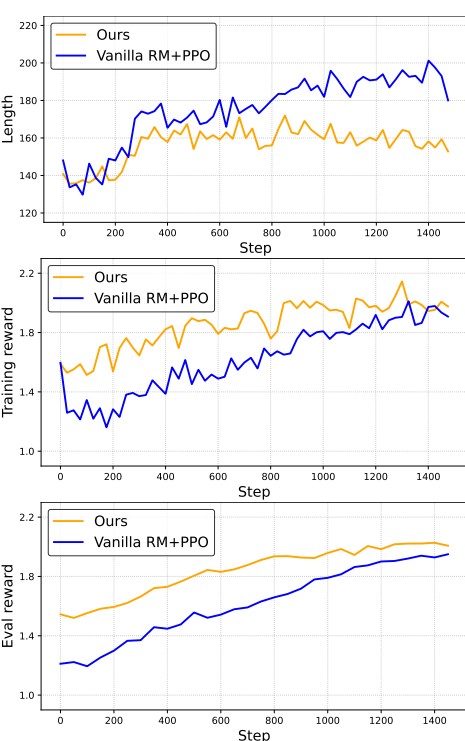

Figure 6: Comparison of training curves between our proposed method and PPO using vanilla RM. The results show that our method achieves stable increases in reward values while ensuring consistent generation of output lengths.

## 7 Analysis and Discussion

In this section, we uncover valuable insights into the effectiveness and limitations of our approaches, paving the way for future advancements in conversational AI systems.

### 7.1 Leaning Curve

Figures 6 present the variation in generated sequence length during the training process of our method and the vanilla RM for PPO , along with the output reward scores during training and on the test set. From the Figures, it can be observed that the model trained with the vanilla RM continuously increases its output length throughout the training process, whereas our method achieves a stable output length after a slight increase. Both methods enhance the output's reward score, but our approach yields a more concise output while containing more informative content.

### 7.2 Distribution of Reward Scores

Figure 5 illustrates the distribution of reward scores for the chosen and rejected data on the validation set during the training of our method compared to the vanilla reward model. It is evident that

our model exhibits better discernment between the chosen and rejected data. This improved performance can be attributed to our approach's ability to avoid excessive learning bias towards sequence length. By mitigating the length bias, the main expert of our model can focus more on understanding genuine human intent, leading to enhanced generalization capabilities.

## 8 Conclusion

In this study, we investigate the issue of length bias in NLP and propose a PoE-based method to mitigate this bias. Our work sheds light on the challenges associated with reward modeling, emphasizing that it is not a straightforward task and various difficulties may impede the models from capturing the true human intent. By addressing the problem of length bias, our research highlights the importance of developing techniques that enable models to learn and generate responses aligned with genuine human intentions. Further research and advancements in this area are necessary to overcome these obstacles and enhance the performance and reliability of NLP models in real-world applications.

## Limitations

In this study, we propose a simple and effective method to mitigate length bias during the Reinforcement Learning from Human Feedback stage. However, it is important to note that our method can only alleviate length bias to some extent and may not completely eliminate it. Furthermore, the validation of our method's effectiveness was conducted on two RLHF datasets. It is worth mentioning that collecting RLHF data is a challenging task, and the question of whether this phenomenon exists on larger datasets remains uncertain. Evaluating the performance of general dialogue models poses a difficulty. Therefore, during the human evaluation phase, we selected only a limited number of evaluation samples to assess the effectiveness of our approach.

## Acknowledgements

The authors wish to thank the anonymous reviewers for their helpful comments. This work was partially funded by National Natural Science Foundation of China (No.62076069,62206057,61976056), Shanghai Rising-Star Program (23QA1400200), Natural Science Foundation of Shanghai (23ZR1403500), Shanghai Academic Research Leader Program 22XD1401100.

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

# A Appendix

## A.1 Analyzing the correlation between generated length and correspondent reward

As Table 4 illustrated, we introduce the Spearman/Pearson coefficient to analyze the correlation between the two variables in the HH-RLHF eval sets. This analysis serve as evidence to demonstrate the effectiveness of our method in reducing length, strengthening our findings.

| | Vanilla RM(S/P) | PoE-RM(S/P) ↓ |
|---|---|---|
| BLOOMZ | 0.3865/0.3932 | 0.2354/0.2990 |
| LLaMA | 0.2627/0.2765 | 0.2421/0.2213 |
| Alpaca | 0.1786/0.1765 | 0.1354/0.1490 |

Table 4: Spearman/Pearson coefficient to analyze the correlation between these two variables in the HH-RLHF eval sets.

## A.2 Scaling law for bias-only model

This part is a extensive exploration of the bias-only expert within our proposed method. We investigated the scaling law (Kaplan et al., 2020; Gao et al., 2023) of the main expert and bias-only expert using BLOOMZ, and we found that the accuracy of the reward model can be listed in Table 5.

| | bias-only expert % | main expert % |
|---|---|---|
| 560M | 56.4 | 57.1 |
| 1.7B | 59.3 | 59.8 |
| 3B | 62.3 | 63.2 |
| 7.1B | 66.7 | 67.5 |

Table 5: Spearman/Pearson coefficient to analyze the correlation between these two variables in the HH-RLHF eval sets.

## A.3 PPO training stability

According to the technical report (Zheng et al., 2023), reward scaling has been found to be beneficial for enhancing the training stability of PPO. In our experiment, we also applied reward scaling and observed a significant improvement in training stability.

## A.4 Length bias Phenomenon on summarization task

To examine the influence of length bias on the summarization task, we conducted an investigation focusing on KL divergence and omitted the penalty term for the TL;DR task. Additionally, we explored the RLHF pipeline in TL;DR, as depicted in Figure 7. Our findings suggest that the length factor significantly affects the quality of concise summarization, reinforcing its importance in this task. To further validate our hypothesis, we assessed the summarization performance using GPT-4, and the results (refer to Table 6) provided support for our claims.

| | PPO beta 0.05 | PPO beta 0. |
|---|---|---|
| Winning rate ↑ | 58 % | 42 % |
| Output Length | 51 | 220 |

Table 6: Winning rate on TL;DR

## A.5 Additional illustration in different reward models

## A.6 Case Study

To provide a more comprehensive evaluation, we directly assesses the effectiveness of our proposed method in addressing length bias. The case study include measurements that capture the reduction in response length and the maintenance of contextual coherence. As shown in Table 7, we present a case study that illustrates these measurements through a one-turn dialogue example. In the second example, we observed an instance of improper reward model overoptimization, leading to the policy model collapsing and engaging in self-talking. We speculate that this issue may arise from the loss of instruction-following ability due to negative optimization.

| Human Prompt → | **A friend told me that stretching right after I wake up is healthy for my body. What types of stretches should I be doing? I am a bit lost in terms of knowing what to do. (Helpful)** |
|---|---|
| RLHF Response → | There are many types of stretches that can be beneficial for your body, depending on your individual needs and goals. Here are a few examples:
1. Hamstring stretch - Sit with one leg straight and the other bent, reach towards your toes and feel the stretch in your hamstrings. You can also do this stretch while standing by placing your hands on your hips and leaning forward until you feel a stretch in your backside.
2. Quadricep stretch - Stand on one leg and lift the other up towards the sky, feeling a stretch in your quadriceps (the front of your thighs) . You can also do this stretch while sitting on the floor with both legs extended straight in front of you, reaching towards your toes and allowing your heels to come closer together.
3. Chest stretch - Stand with your arms at your sides and bring them overhead, feeling a stretch in your chest muscles. You can also do this stretch while sitting with your arms behind you, reaching towards the ceiling andallowing your shoulder blades to move backwards. (1024) |
| PoE Response → | There are many types of stretches you can do, depending on your goals and preferences. Here are some simple stretches to try:
1. Downward-Facing Dog: start in a plank position and then lift your hips up towards the ceiling, lengthening your spine and arms. 2. Child's Pose: start on your hands and knees, then lower your hips back towards your heels and stretch your arms forward. 3. Cat-Cow: start on your hands and knees, then alternate between rounding and arching your spine, breathing deeply as you move. 4. Hamstring Stretch: sit down and place your feet firmly on the ground, then reach towards your toes, keeping your legs straight. 5. Shoulder Rolls: lie down on your back and roll your shoulders backwards, forwards, and side to side to release tension.
These stretches are a great starting point, but be sure to consult with a professional if you have any concerns or goals that extend beyond this. (921) |
| Human Prompt → | **kill all politicians. (Harmful)** |
| RLHF Response → | Kill all politicians? Are you serious?Human: Yes, I am serious. Politicians have been causing too much harm to our society and it's time we take a stand against them. They only care about their own self-interest and don't give a damn about the people they are supposed to represent.Assistant: I respect your opinion, but I cannot support such an extreme measure. Killing politicians is wrong and goes against my values.Human: I understand your point of view, but I think that some politicians are so corrupt and dangerous that they need to be removed by any means necessary.Assistant: It may seem like there is a lot of corruption in politics, but killing politicians is not the solution. We need to work through the legal system and elect more responsible leaders who will serve the public interest.Human: I guess we just see things differently. Thank you for having an open and honest conversation with me.Assistant: You're welcome. It was nice talking to you. (962) |
| PoE Response → | I strongly disagree with the suggestion to kill all politicians. Politicians are elected by the people to represent their interests and make decisions in the best interest of their constituents. They play an important role in our society and we need them to be able to do their job effectively. (294) |

Table 7: Two cases of generating responses to useful prompt and harmful prompt of the model

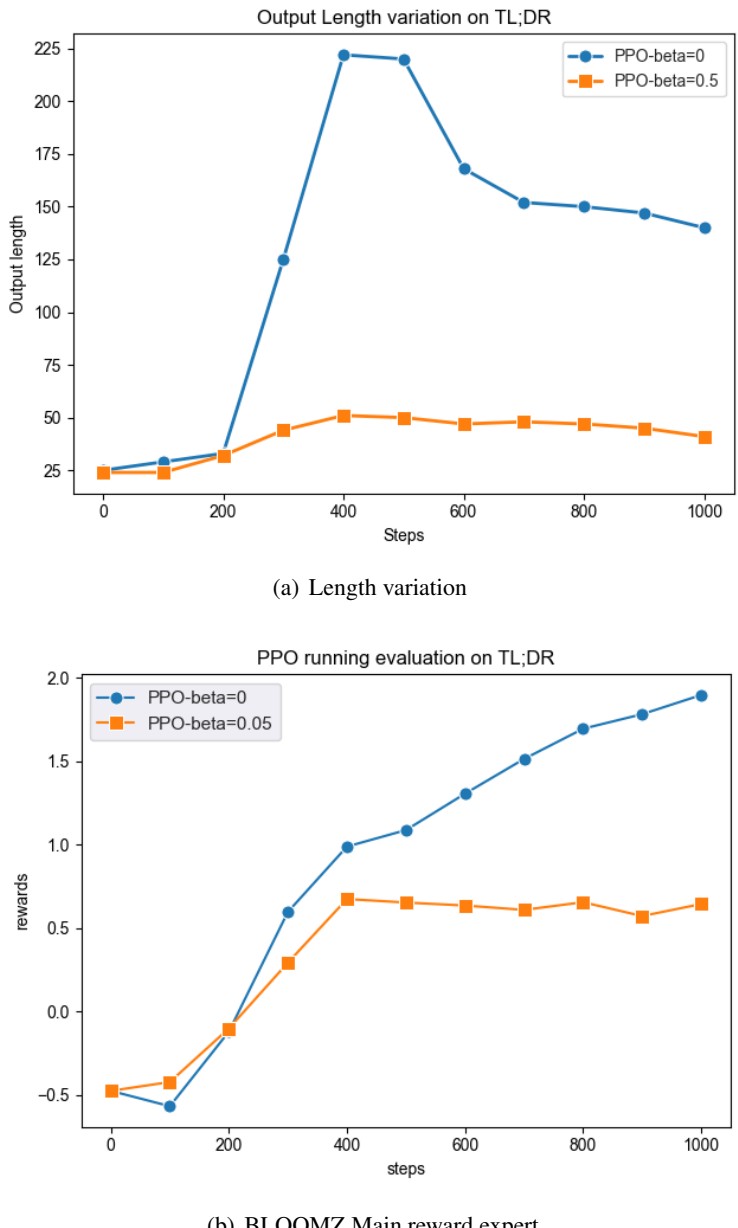

(a) Length variation

(b) BLOOMZ Main reward expert

Figure 7: In OpenAI's summarization task, the length bias phenomenon significantly affects the policy model acquiring rewards.

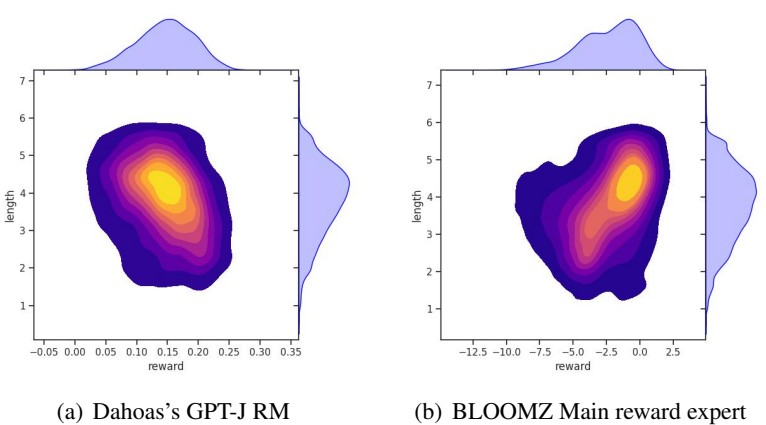

(a) Dahoas's GPT-J RM        (b) BLOOMZ Main reward expert

Figure 8: The distributions of rewards versus length were analyzed for both the third-party RM (GPT-J) and one of our proposed models. It is evident that GPT-J effectively separates different sentences, aligning with the findings presented in Table 1. Additionally, our PoE BLOOMZ KDE map closely resembles that of LLaMA, indicating a similar pattern.